# Genetic Diversity and Population Structure of Local Chicken Ecotypes in Burkina Faso Using Microsatellite Markers

**DOI:** 10.3390/genes13091523

**Published:** 2022-08-24

**Authors:** Zare Yacouba, Houaga Isidore, Kere Michel, Gnanda B. Isidore, Traore Boureima, Millogo Vinsoun, Konkobo Maurice, Bandaogo Ousseni, Zongo Moussa, Bougouma-Yameogo M. C. Valerie, Rekaya Romdhane, Nianogo A. Joseph

**Affiliations:** 1Laboratoire de Recherche en Production et Santé Animales (LaRePSA), Institut de l’Environnement et de Recherches Agricoles (INERA), Ouagadougou 01 BP 476, Burkina Faso; 2Institut du Développement Rural, Université Nazi BONI, Bobo-Dioulasso 01 BP 1091, Burkina Faso; 3Centre for Tropical Livestock Genetics and Health (CTLGH), Roslin Institute, University of Edinburgh, Easter Bush Campus, Edinburgh EH25 9RG, UK; 4Laboratoire de Physiologie Animale, Unité de Formation et de Recherche en Sciences de la Vie et de la Terre (UFR/SVT), Université JosephKI-ZERBO, Ouagadougou 03 BP 7021, Burkina Faso; 5Centre Universitaire de DORI, Université Thomas SANKARA, Ouagadougou 12 BP 417, Burkina Faso; 6Centre International de Recherche-Développement sur l’Elevage en zone Subhumide (CIRDES), Bobo-Dioulasso 01 BP 454, Burkina Faso; 7Institut National de Santé Publique, Centre MURAZ, Bobo-Dioulasso 01 BP 390, Burkina Faso; 8Department of Animal and Dairy Science, Department of Statistics, Institute of Bioinformatics, 106, University of Georgia, Athens, GA 30602, USA

**Keywords:** local chicken, Konde chicken, genetic diversity, microsatellite markers, Burkina Faso

## Abstract

The objective of this study was to investigate the genetic diversity and population structure of local chicken ecotypes from Burkina Faso using microsatellite markers. A total of 71 individuals representing local chicken populations from the Centre-East (18), Centre-North (17), Sahel (18) and South-West (18) were used to estimate genetic diversity indices, population structure and phylogenetic relationships using 20 selected polymorphic microsatellite markers. The number of alleles, mean number of alleles, mean of observed and expected heterozygosity and polymorphic information content were 127, 6.35, 0.391, 0.521, 0.539 and 0.541, respectively. The estimated overall fixation index between loci (F), among populations (F_IS_) and inbreeding coefficient within chicken ecotypes were 0.239, 0.267 and 0.243, respectively. Analysis of the molecular variance revealed that 77% of the total genetic diversity was attributed to within-population variation and the remaining 1% and 22% were attributed to among-regions differentiation (F_ST_) and among-individual differentiation (F_IT_), respectively. The highest pairwise genetic distance (0.026) was found between the local Konde ecotype and those from the Centre-North region while the lowest distance was observed between local chickens from the Sahel and the Centre-North regions (0.003). Neighbour-joining phylogenetic tree and principal component discriminant analyses confirmed the observed genetic distances between populations. The results show that local chickens in Burkina Faso have a rich genetic diversity with little differentiation between the studied populations. This study provides important information on measures of genetic diversity that could help in the design and implementation of future genetic improvement and conservation programs for local chickens in Burkina Faso.

## 1. Introduction

Genetic diversity, a vital resource for the continuous genetic improvement of livestock and poultry species, has received much attention in recent years [1]. Most local chickens were facing extinction threats when international commercial strains became more common [2]. These introduced chickens have certainly contributed to the dilution of the genetic diversity in local chicken populations and could pose a threat to the existing genetic variability. Local chicken farming in Burkina Faso, as in many developing countries, plays an important role among small-scale producers as the major source of animal protein to humans, income and socio-cultural role [3,4,5,6]. However, in Burkina Faso, studies of the genetic diversity of local chicken populations are scarce and often limited to the morpho-biometric characteristics of these populations [7,8] and to farm typologies [6]. To date, there is no relevant information on the genetic diversity and genetic structure of local chicken population in Burkina Faso. In other words, there is no molecular evidence on whether local chicken ecotypes in Burkina Faso represent genetically distinct populations. Particularly, there is no studies on the molecular characterisation of the endangered local chicken ecotype “Konde” from the Centre-East region of Burkina Faso [9]. In addition, the agro-ecological zone is suspected to affect animal species genetic diversity and distribution [5,7,8]. Therefore, it is of particular interest to identify genetic variability among local chicken ecotypes in Burkina Faso. Molecular analysis using microsatellite markers is an appropriate method to study genetic diversities within and between populations because they are highly polymorphic, exhibit co-dominant inheritance, abundant and homogeneously distributed in the genome [10]. So far, many studies have been conducted to assess the genetic diversity of chickens using microsatellite markers, and the reported results clearly proved the usefulness of these panels for biodiversity and genetic improvement studies [1,2,5,11,12]. Knowledge of the genetic diversity and population structure of local chickens in Burkina Faso can provide a better understanding of the differences and similarities between different populations and serve as a basis for future genetic improvement and the implementation of effective conservation programs. The primary aim of this study was to investigate the genetic diversity, genetic relationships and population structure of local chicken ecotypes from four regions of Burkina Faso using 20 polymorphic microsatellite markers.

## 2. Materials and Methods

### 2.1. Sample Collection and DNA Extraction

Blood samples were obtained from 71 local chickens from four regions of Burkina Faso: Centre-East (N = 18), Centre-North (N = 17), Sahel (N = 18) and South-West (N = 18). One bird was chosen from each of the participating 71 farms. Chicken blood was sampled from farms in at least five communes per region. The description of the local chicken populations has been reported previously [7]. Populations were inferred based on agro-ecological zone and phenotypes [7]. Whole blood was collected by bleeding the wing vein. Blood was then transferred to serum tubes containing EDTA or heparin (anticoagulants) and stored at 8 °C. The QiAGen kit was used for DNA extraction. DNA concentration was measured using a spectrophotometer (NanoDrop 2000c—Thermo Scientific) (Thermo Fisher Scientific, Wilmington, NC, USA). The DNA was then stored at −20 °C for further molecular applications. 

### 2.2. DNA Amplification and Microsatellite Genotyping

Twenty microsatellite markers were used to assess DNA polymorphism. All markers were among the 30 microsatellites recommended by the International Society of Animal Genetics (ISAG)-FAO for polymorphism assessment [13]. Polymerase chain reaction (PCR) amplifications were performed based on simplex PCR techniques using QIAGEN multiplex hot reactors using QIAGEN hot star enzyme (QIAGEN, Valencia, CA, USA) and the multiplexes were manually optimised. The 20 microsatellite markers were pooled into three multiplexes each with 6–7 primer pairs per reaction plate. A final volume of 10 μL of simplex PCR mix multiplex PCR master mix contained: 10x PCR buffer with MgCl_2_ (25 mM), dNTPs (2.5 mM), QIAGEN HotStar Tag (5 U/μL), distilled water, direct and reverse non-fluorescently labelled primer pairs (Forward and Reverse) each with a concentration of 10μM and a fluochrome consisting of a FAM, VIC, NED and PET-labelled M13 sequence and genomic DNA (25 ng/μL). Amplification using the BIOMETRAABI9700 thermal cycler (BIOLABO Scientific, Archamps, France) was performed according to the following program: an initial denaturation at 94 °C for 15 min, followed by 35 cycles (94 °C, 30 s; T° hybridisation of primers for 45 s and 72 °C, 1 min) and a final extension of 72 °C for 10 min. After amplification, plate rearrangement consisted of multiplexing, i.e., mixing of several microsatellite markers that had been amplified by simplex PCR. This required an adjustment to balance the reaction medium so that all the alleles of the different mixed microsatellite markers can be distinguished. Fragment analysis was performed on an ABI capillary sequencer the Genetic Analyzer 3500 (Applied BioSystems or ABI 3500) (Thermo Fisher Scientific, Tokyo, Japan). Analysis of migration profiles was performed using GeneMapper 5 software (Applied BioSystems) (Thermo Fisher Scientific, Tokyo, Japan).which is a tool developed for reading migrated microsatellite profiles on the ABI 3500 (Thermo Fisher Scientific, Tokyo, Japan) was used to determine fragment size and allele identification by comparing them to a known internal size standard.

### 2.3. Estimates of Genetic Diversity and Distance

Basic measures of inter- and intra-population genetic diversity, such as total number of alleles, number of effective allele (Ne), allelic frequencies, average number of alleles (Na), observed (Ho) and expected (He) number of alleles, polymorphism information content (PIC), Shannonindex (I),fixation index (F) and Wright’s F-statistics (F_IT_, F_ST_ and F_IS_) were calculated according to the methods presented in Weir [14] using FSTAT2.9.4 (Jerome Goudet, University of Lausanne, Swiss) and GenAlEx ver.6.5software (Peakall and Smouse respectively in Australia and USA).

### 2.4. Genetic Relationships and Structure 

The genetic relationships between local chicken ecotypes in Burkina Faso were studied using two methods. In the first method, we bootstrapped 1000 times on all loci using DARWIN6 software (CIRAD-BIOS Departement, Montpellier, France) to establish the phylogenetic tree and dendrogram of the local chicken ecotypes in Burkina Faso using the standard genetic distance of Nei [15] and the neighbour-joining method. In the second method, the genetic structure of the studied chicken ecotypes was inferred from the multi-loci genotypic data using a Bayesian approach employed in STRUCTURE software (Pritchard, J. K., Stephens, M., and Donnelly, P.; University of Oxford) version 2.3.4 [16], STRUCTURE Harvester (University of California, USA) [17] and CLUMPAK (Naama M Kopelman, Jonathan Mayzel; Tel Aviv University, Israel). The analysis was performed using an admixture model with independent allele frequencies between ecotypes [16,18]. The STRUCTURE analysis was implemented using a unique Markov Chain Monte Carlo (MCMC) chain of 120,000 rounds where the initial 20,000 were discarded as burn-in period and K (number of clusters) ranging from 2 to 10. For each value of K, 100 independent runs were performed. The most probable number of clusters (ΔK) was calculated following the equations proposed by Evanno et al. [19]. Principal component discriminant analysis (PCDA) was performed using the method implemented in the ADEGENET package [20] within the R (George Ross IhakaandRobert Clifford Gentleman respectively in New Zealand and Canada) statistical package version 4.2.1 (2022).

## 3. Results

### 3.1. Estimates of Genetic Diversity

In this study, 127 alleles were detected across the 20 polymorphic markers (Table 1). The 20 microsatellite markers had an average of 6.35 alleles (Na) with a mean effective number of alleles (Ne) of 2.304 where the averages of Ho, He, uHe and polymorphism information content (PIC) were 0.391, 0.521, 0.539 and 0.541, respectively. Marker LEI0192 had the highest number of alleles (18) while markers MCW0103 and MCW0098 each had the lowest number of alleles (2). The locus PIC values ranged from 0.103 (MCW0248) to 0.773 (LEI0192) among the loci studied (Table 1). 

On the other hand, considering the four local chicken ecotypes, the highest Shannon information index, an estimator of the diversity index, ranged between 0.894 ± 0.083 and 1.054 ± 0.086 in Sahel and South-West chickens, respectively (Table 2). The mean Ho was 0.391 ± 0.022 among the study populations. However, the highest Ho was found in Konde ecotype chickens (0.450 ± 0.041) and the lowest heterozygosity (0.364 ± 0.047) corresponded to chickens from the Sahel region. On the other hand, the unbiased mean He of Nei was 0.539 ± 0.018 for all loci and ranged from 0.506 ± 0.036 to 0.567 ± 0.037 in local chickens from the Sahel region and the South-West region, respectively (Table 2). The estimated overall fixation index (F) was 0.239 ± 0.037 for all loci with a range between 0.106 ± 0.057 (Konde chicken) and 0.324 ± 0.076 (South-West).

### 3.2. Genetic Distances of Local Chicken Ecotypes

The pairwise estimates of the distances (F_ST_) between the studied populations ranged between 0.003 and 0.026 (Table 3). The highest pairwise genetic distance (0.026) was found between the local Konde ecotype chickens and those from the Centre-North region while the lowest distance was observed between the local chickens from the Sahel and Centre-North regions (0.003). The results of Nei’s unbiased genetic distance were also in agreement with the pairwise genetic distances showing the highest distance (0.032) between local Konde ecotype chickens from the Centre-East region and those from the Centre-North region, while the lowest genetic distance (0.002) was between local chickens from the Sahel and Centre-Northregions.

The pairwise genetic distance matrix between ecotypes showed no significant genetic distance between ecotypes from the Sahel and Centre-North regions. In contrast, the genetic distance between all other groups was significant (*p* < 0.05). Analysis of molecular variance revealed that there was only 1% variation between local Burkina Faso chicken ecotypes and that 22% of the variation is due to differentiation of individuals in relation to the total ecotypes. The greatest variability (77%) was due to within individual variation that was represented by diversity within the population with an overall F_IS_ value of 0.235 (*p* < 0.001) (Table 3 and Table 4). Based on Weir [14] proposed estimator, the average inbreeding within the population (F_IS_) was 0.243 for the 20 loci and ranged from −0.136 (MCW locus 0067) to 0.774 (MCW locus 0098) (Table 1 and Table 3). The overall heterozygous deficit or total inbreeding (F_IT_) ranged from −0.123 (MCW0103) to 0.787 (MCW0098 locus) with a mean of 0.267. The mean genetic distance (F_ST_) was 0.036 for the 20 loci and ranged from 0.004 (MCW0103) to 0.120 (MCW0216).

### 3.3. Phylogenetic Analysis and Cluster of Local Chicken Ecotypes

The evolution of K value and the unrooted consensus tree obtained using the STRUCTURE software revealed two main groups (Figure 1 and Figure 2), with Konde ecotype from the Centre-East and local chickens from the South-West region in one group, and local chicken ecotypes from the Sahel and Centre-Northregions in the second group. This clustering of local chicken ecotypes into two large groups revealed the presence of a clear genetic separation between the ecotypes of the different groups.

The dendrogram (Figure 3), the neighbourhood phylogenetic tree (Figure 4), STRUCTURE analysis (Figure 5)divided the populations into three different subgroups with a high level of genetic mixing and gene flow between them. The structure program was used to study the genetic structure of local chickens in the four regions of Burkina Faso. When the value of K (number of clusters) was low (i.e., K = 2), individuals were grouped into putative populations in a manner similar to the results presented in the neighbour-joining tree. When K was equal to 3, the local Konde chicken ecotype from the Centre-East and the local chicken ecotype from the South-West region clustered independently, and thus could be considered genetically distinct subpopulations. The local chicken ecotypes from the Sahel and Centre-Northregions did not show in separate clusters. However, when K was set equal to 4, the four local chicken ecotypes were placed in separate clusters, as expected (Figure 2). The average log-likelihood of the data increased steadily from K = 2 to K = 4 and showed a plateau-like appearance with no significant change from K = 3 to K = 4 (Figure 1). The results showed better agreement in the structure output for values of K between 2 and 4 (Figure 1). The output at K = 3 appeared to be the best; it clearly distinguished the different chicken population. The output of the Evanno table describing the population structure parameters is presented in Figure 1 illustrating the evolution of the average estimate of the delta K (estimate of the most likely number of groups in the local chicken population of Burkina Faso). The results show that the most probable number of clusters is three. In agreement with the Nei distances and Wright’s coefficients, the dendrogram (Figure 3) and the phylogenetic relationship by the neighbour-joining tree (Figure 4)show that the local chickens of the Konde ecotype tend to be found in one cluster (I). Only two individuals are found in the second group (II) and no individuals of the local Konde ecotype in the third group (III). However, several individuals of the local chicken ecotypes from the Sahel and Centre-North regions were found in all groups (Figure 3, Figure 4 and Figure 5). 

The DAPC (Figure 6) showed little genetic distance between the subpopulations.Also a high gene flow revealed between the different populations (Figure 3, Figure 4, Figure 5 and Figure 6). Among the three groups defined at K = 3 (Figure 5), Cluster 1 (red) consisted predominantly of birds from the Centre-East (88.89%), Centre-North (41.18%), Sahel (61.11%) and South-West (50%). The second cluster (green) included 35.29, 27.78 and 33.33% of birds in the Centre-North, Sahel and South-West. No bird from the Konde ecotype were included in this cluster. The third cluster (blue) included 11.11, 23.53, 11.11 and 16.67% of individuals from the Centre-East (Konde ecotypes), Centre-North, Sahel and South-West, respectively. This structuring shows the same trend as the phylogeny tree and dendrogram results.

## 4. Discussion

### 4.1. Allelic Diversity

The mean number of alleles (6.35) found in our study is slightly lower than the estimates obtained in five agro-ecological zones of Cameroon (7.09) [12], using local chickens of Cameroon (9.04) [5] and using four subpopulations of Aseel chickens in Pakistan [21]. However, our estimated mean number of alleles was markedly lower than estimates of 10.33 [22], 14.17 [23] and 16.8 [24] using Indian, Bhutanese and Chinese chicken populations, respectively. Further, our estimate of the mean number of alleles was higher than those reported in Kenya (1.96) [11], using five local breeds of chicken in Sweden (4.7) [2] and using three breeds of chicken in India (4.8) [25]. Our estimate was close to those obtained in Tanzania (5.10 to 6.28) [26] and using local Ethiopian chickens (6.5) [27]. The estimated mean number of effective allele in this study (2.304) was lower than estimates obtained in Pakistan (6.0) [21], in India (3.09) [25], in Cameroon (3.13) [5] and in China (4.8) [24]. Lower estimates than ours were reported using native chickens in Kenya (1.726) [11]. This variation in the average number of alleles per locus and the actual number of alleles may be related to the number or type of markers used, the sample size and the genetic resource studied.

### 4.2. Heterozygosity Rate

This study showed that the observed mean heterozygosity of the different chicken populations for the 20 microsatellite loci ranged from 0.078 to 0.635 while the expected and unbiased heterozygosity ranged from 0.101 to 0.727 and 0.104 to 0.750, respectively. Heterozygosity is a well-known measure of genetic diversity. The average Ho, He and uHe values between populations were0.391 (ranged from 0.364 to 0.450), 0.521 (ranged from 0.489 to 0.547) and 0.539 (ranged from 0.506 to 0.567), respectively. Genetic diversity at loci observed in local chicken ecotypes from Burkina Faso is lower than those reported in 52 populations across 22 loci (0.47) [28], in local chickens from Cameroon (0.60) [5], in six breeds of Mediterranean chickens (0.46) [29] and in native chicken population from Bangladesh (0.67) [30]. Additionally reported were higher values for Ho (0.71 to 0.88) [31] and He (0.47 to 0.85) using Korean native chicken lines. In contrast, our results were higher than those obtained in Kenyan native chickens (He = 0.40) [11] and in a population of four local breeds of Swedish chickens (0.32) [2]. The average population heterozygosity reflects the degree of population homogeneity [32]. The higher the genetic similarity in a population, the lowerits average heterozygosity, and the opposite is true. Variation in observed and expected heterozygosity can be attributed to differences in location, sample size, population structure and microsatellite marker sources [33].

### 4.3. Polymarphism Informative Content

According to some authors [34,35], loci are highly informative when their PIC is greater than 0.5. A PIC between 0.25 and 0.5 indicates a reasonably informative locus. Marker with a PIC smaller than 0.25 are only slightly informative. Thus, 80% of the loci used in this study were highly informative with the LEI 0192 (PIC = 0.77) being the most informative. The average information content was 0.541 indicating that the microsatellite loci selected in this study were reasonably informative and were appropriate to assess the genetic diversity of the chicken populations in Burkina Faso. The PIC values obtained in this study were comparable to those reported [36] and higher than the estimates obtained using 11 local Chinese chickens populations [37]. However, our estimates were lower than those reported using Pakistani [21], Korean [31] and Bangladeshi [30] native chicken populations, respectively. In fact, their reported estimates of PIC ranged between 0.60 and 0.87.

### 4.4. Genetic Distance

Estimates of the genetic distance clearly indicate that the main source of genetic diversity between the studied local chicken ecotypes is between individual variation within population. Analysis of molecular variability revealed that the genetic distance between local chickens in the Centre-East (Konde ecotype), Centre-North, Sahel and South-West regions of Burkina Faso was very narrow, and it was largely due to differences within individuals.small genetic distance was reported in local chicken populations from northwestern Ethiopia (0.07 to 0.13) [27] and Kenya (0.02 to 0.13) [11], respectively.

On the other hand, our estimates of genetic diversity were lower than those found for native Bangladeshi chickens (0.29 to 0.58) [30], and five Korean chicken lines (0.08 to 0.17) [31]. These results imply that the overall diversity is mainly due to the diversity among individuals within population. The Wright’s F-statistic(FST) was equal to 0.036 and was much lower than the estimates reported for Aseel chickens [21], but similar to the estimates obtained for several local chicken ecotypes and populations in Cameroon [5] and Kenya [11], respectively.

### 4.5. Phylogenetic Analysis

Nei’s unbiased genetic distance matrices and dendrogram showed that local chickens from the Sahel and North-Central regions were grouped in the same branch. They formed a distinct group with the same origin that revealed their close relationships. The local Konde chickens from the Centre-East region and the South-West region were separated by a node, which means that the Konde chickens constitute a distinct chicken subpopulation with a common ancestor. This clustering results could also be explained by geographic realities, as chickens in the Sahel and Centre-Northregions trace their origin to chickens from neighbouring regions. Konde ecotype chickens and local chickens from the South-West region may have been separated from their common origin in a recent past. The large mixture of local chickens reflected in the phylogenetic tree reflects the high proportion of shared alleles that results from high gene flow between the four ecotypes. The population structure of local chickens in Burkina Faso, characterised by a high level of admixture, has also been observed in other local African chicken populations such as the case in Kenyan [11], Cameroonian [5] and Zimbabwean [38]. Similar results were observed in Asian local chicken populations [39]. This type of structuring reflects low differentiation and high levels of gene flow. According to Nei’s standard genetic distance, this small level of differentiation can be due to mutations and genetic drift. However, the Reynolds distance is only caused by the genetic drift [15]. The sampled chickens were selected from farms based on phenotypic information from the randomly mated population. The high diversity observed in local chicken ecotypes and the low genetic differentiation between local ecotypes in Burkina Faso are certainly due to the breeding system characterised by uncontrolled breeding and movement of chickens from one region to another, thus favouring a permanent gene flow. The absence of organised selection programs with clear objectives and the uncontrolled migration of chickens between farms and regions have caused a continuous gene flow between ecotypes.

## 5. Conclusions

This study of local chicken ecotypes from the Centre-East (Konde chicken), Centre-North, Sahel and South-West regions of Burkina Faso based on 20 informative microsatellite loci clearly demonstrated the genetic diversity observed at the phenotypic level. The analysis of the structure of the studied ecotypes revealed three sub-populations. The Konde ecotype chicken is a distinct sub-population with likely common ancestor with chickens from the South-West region, while the local chicken ecotypes of the Centre-Northand Sahel regions form a single population. This study provides important information on the genetic background of local chicken genetic resources in Burkina Faso that could be used for conservation and in potential genetic improvement programs.

## Figures and Tables

**Figure 1 genes-13-01523-f001:**
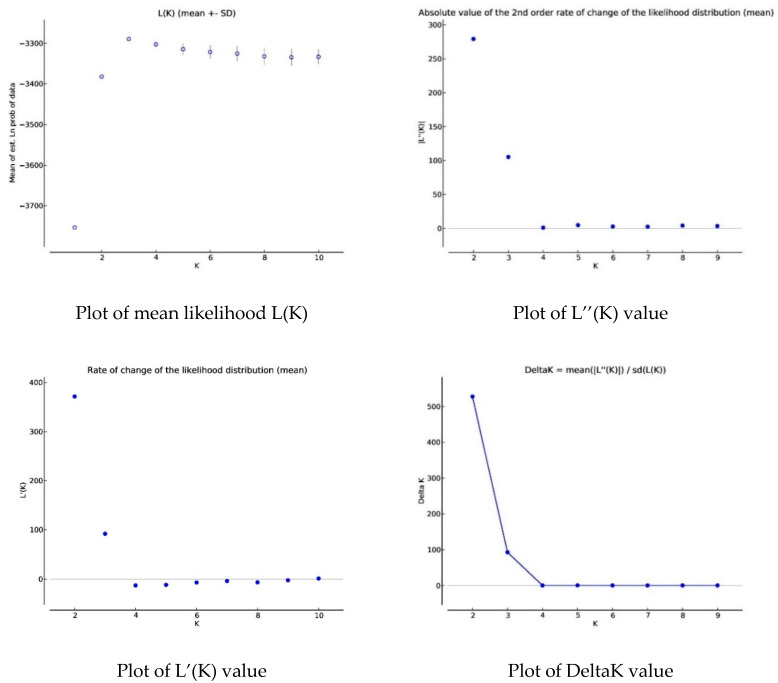
Evolution of the parameters estimating the value of K (likelihood).

**Figure 2 genes-13-01523-f002:**
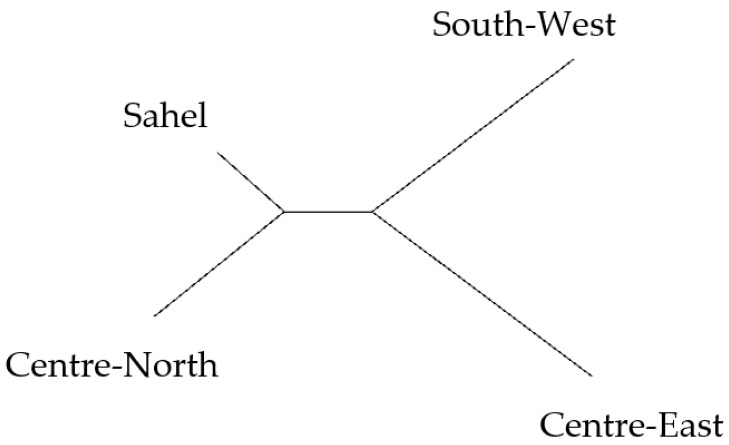
Unrooted consensus tree constructed using standard genetic distance (Nei, 1972) describing the relationships between local chickens in the Centre-East (Konde ecotype), Centre-North, Sahel and South-West regions of Burkina Faso.

**Figure 3 genes-13-01523-f003:**
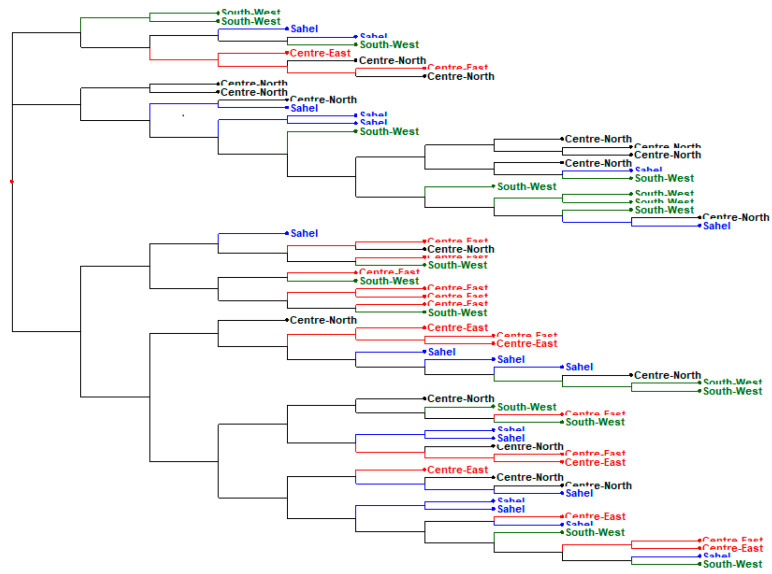
Dendrogram obtained by neighbour-joining cluster analysis among four local chicken ecotype in four regions evaluated using 20 microsatellites markers.

**Figure 4 genes-13-01523-f004:**
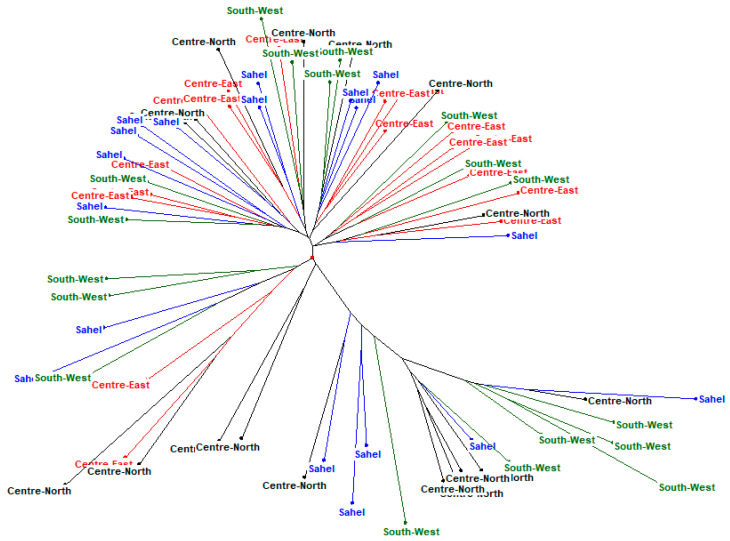
Phylogenetic tree based on the neighbour-joining describing the relationships between local chickens in the Centre-East (Konde ecotype), Centre-North, Sahel and South-West regions of Burkina Faso.

**Figure 5 genes-13-01523-f005:**
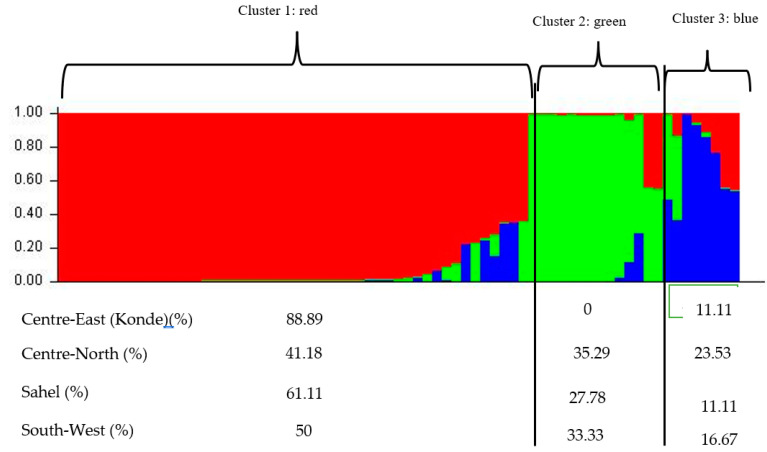
Genetic structure of four local chicken ecotypes from the Centre-East (Konde ecotype), Centre-North, Sahel and South-West regions of Burkina Faso.

**Figure 6 genes-13-01523-f006:**
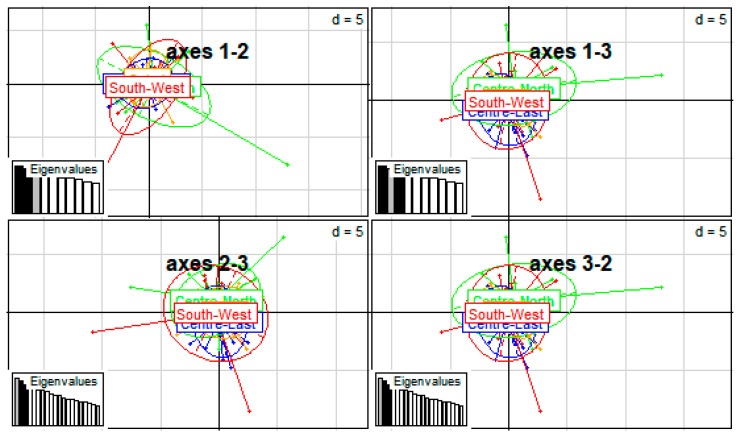
Graphical representation of the discriminant principal component analysis (DAPC) using four local chicken ecotypes from the Centre-East (ecotype Konde), Centre-North, Sahel and South-West regions of Burkina Faso for spatial distribution.

**Table 1 genes-13-01523-t001:** Number of alleles (Na), polymorphism information content (PIC), observed (Ho) and expected (He) and unbiased (uHe) heterozygosity, fixation index (F), Wright’s F-statistics (F_IS_, F_IT_ and F_ST_) and Hardy Weinberg (HW) tests.

	Na	PIC	Ho	He	uHe	F	F_IS_	F_IT_	Fst	Prob	Signif (HW)
ADL0268	8	0.633	0.593	0.619	0.64	0.038	0.042	0.064	0.023	0	***
MCW0206	6	0.618	0.635	0.603	0.621	−0.053	−0.052	−0.027	0.024	0	***
ADL0278	7	0.7	0.516	0.676	0.699	0.225	0.236	0.262	0.035	0	***
MCW0103	2	0.297	0.333	0.296	0.304	−0.133	−0.127	−0.123	0.004	0.305	ns
MCW0037	3	0.549	0.336	0.511	0.53	0.35	0.341	0.387	0.069	0.009	**
MCW0183	11	0.602	0.516	0.594	0.611	0.125	0.132	0.144	0.013	0	***
MCW0069	6	0.562	0.449	0.554	0.571	0.178	0.189	0.201	0.014	0	***
MCW0081	8	0.578	0.595	0.569	0.588	−0.052	−0.046	−0.031	0.015	0	***
MCW0222	5	0.553	0.172	0.531	0.552	0.679	0.676	0.689	0.04	0	***
MCW0216	4	0.605	0.276	0.533	0.552	0.467	0.481	0.544	0.12	0	***
MCW0098	2	0.391	0.083	0.368	0.383	0.831	0.774	0.787	0.057	0	***
MCW0111	6	0.734	0.37	0.691	0.717	0.469	0.465	0.496	0.059	0	***
MCW0330	7	0.541	0.22	0.535	0.558	0.595	0.59	0.594	0.01	0	***
MCW0067	4	0.31	0.343	0.302	0.314	−0.115	−0.136	−0.106	0.026	0.664	ns
MCW0295	7	0.548	0.589	0.529	0.546	−0.116	−0.113	−0.075	0.034	0.996	ns
MCW0248	3	0.103	0.078	0.101	0.104	0.125	0.222	0.237	0.019	0.116	ns
ADL0112	6	0.637	0.467	0.615	0.635	0.25	0.241	0.267	0.035	0	***
MCW0034	10	0.512	0.319	0.496	0.512	0.34	0.356	0.376	0.03	0	***
MCW0078	4	0.588	0.348	0.572	0.59	0.396	0.391	0.408	0.028	0	***
LEI0192	18	0.773	0.583	0.727	0.75	0.189	0.198	0.245	0.059	0	***
Mean	**6.35**	0.541	0.391	0.521	0.539	0.239	0.243	0.267	0.036		

ns: not significant; ** *p* < 0.01, *** *p* < 0.001.

**Table 2 genes-13-01523-t002:** Effective number of alleles (Ne), Shannon index (I), observed (Ho), expected (He) and unbiased (uHe) heterozygosity, by ecotype.

Population		Ne	I	Ho	He	uHe
**Centre-East** **(Konde)**	**Mean**	2.246	0.938	0.45	0.518	0.533
	**SE**	0.13	0.069	0.041	0.036	0.037
**Centre-North**	**Mean**	2.356	1.008	0.367	0.53	0.55
	**SE**	0.165	0.086	0.042	0.037	0.038
**Sahel**	**Mean**	2.161	0.894	0.364	0.489	0.506
	**SE**	0.183	0.083	0.047	0.035	0.036
**South-West**	**Mean**	2.454	1.054	0.384	0.547	0.567
	**SE**	0.173	0.086	0.047	0.036	0.037
**Total**	**Mean**	2.304	0.973	0.391	0.521	0.539
	**SE**	0.081	0.04	0.022	0.018	0.018

SE: Standard error.

**Table 3 genes-13-01523-t003:** Population pairwise F_ST_ (below the diagonal) and Nei’s unbiased genetic distance (above the diagonal) in local chickens from the Centre-East (Konde ecotype), Centre-North, Sahel and South-West regions of Burkina Faso.

	Centre-East	Centre-North	Sahel	South-West
**Centre-East**	0.000	0.032	0.020	0.026
**Centre-North**	0.026 **	0.000	0.002	0.020
**Sahel**	0.018 *	0.003 ^ns^	0.000	0.021
**South-West**	0.021 **	0.016 **	0.020 **	0.000

** *p* < 0.01; * *p* < 0.05; ns = non significant.

**Table 4 genes-13-01523-t004:** Analysis of molecular variance calculated on the basis of the allelic distance matrix of Wright’s F-statistics among local chickens in the Centre-East (Konde ecotype), Centre-North, Sahel and South-West regions of Burkina Faso.

Source	Degree of Freedom	Sum Square	Mean Square	Estimated Variances	% of Estimated Variances	F-Statistics Value	*p* (Rand ≥ Data)
**Among ecotype**	3	26.39	8.80	0.06	1	F_ST_ = 0.011	0.005
**Among populations within ecotype**	67	445.82	6.65	1.23	22	F_IT_ = 0.227	0.000
**Within Individual**	71	297.67	4.19	4.19	77	F_IS_ = 0.235	0.000
**Total**	141	769.88		5.48	100		

## Data Availability

Not applicable.

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
