# Peer review of "Genetic Diversity and Population Structure of Local Chicken Ecotypes in Burkina Faso Using Microsatellite Markers"

_genes, 2022, doi:10.3390/genes13091523_

Round 1
Reviewer 1 Report
This submission studied the genetic diversity and population structure of 4 chicken populations local to Burkina Faso.
While the manuscript gives valuable insights into these populations, the manuscript lacks polish and consistency.
Examples include (in most cases the grammatical suggestions apply in several places throughout the manuscript):
The comma in the last sentence in the Abstract is not needed, but one is needed in the compound sentence in the Introduction, “So far, many studies…microsatellite markers, and the reported results…”
In the Introduction, “Particularly, studies on the molecular…” is not a complete sentence. Studies (the noun) needs a verb.
Materials and Methods 2.1 – Chicken blood (s not needed); brachial vein?
2.2 – “After amplification, plate rearrangement consists…” The tense changed from past tense to present for 2 sentences.
2.3 – The list of measures needs a comma before the last measure.
Results 3.1 – Shouldn’t Ho, He, and uHe be defined before using their abbreviation, just as you did for PIC?
LEI0192 or LEI 0192?
Shannon Index needs another “n” throughout the manuscript.
I believe this is the first time Konde is introduced, but no reference is made here or in Table 2 to Centre-East to know they are the same.
3.2 – Heading is italics
The sentence, “The greatest variability (77%)…” doesn’t make sense, because it is referring to within populations, but then jumps to among populations (Fst) without a transition.
Table 1 – N instead of Na?, PIC is listed twice. Is the bottom row total or average?
Tables 1 and 2 – Commas to denote decimals when the text and following Tables use decimal points. They are inconsistent.
Table 2 – It would be more clear if the measures were listed in the same order in the title as in the table. What are Ne and I? Neither is defined in the title.
Table 4. Should the title state “Wright’s F-statistics” as in M & M instead of FST, because the table lists all three?
Based on the left-most column, Fis and Fit are switched. The table states that Fis is among individuals and Fit is within individuals. However, the Abstract states that Fis is within and Fit is between. Also, in the Abstract, it should state “…heterozygote deficiency within (Fis) and between (Fit) chickens in the populations…”. Otherwise, Fit is stated the same as Fst (between populations).
Figure 1 would be more clear if each quadrant of the figure had a title.
Figures 2, 3, and 5 are not in English. The text, particularly in Figure 3 is also very small and hard to read.
Throughout, the way Fit, Fis, and Fst are written is inconsistent (capitalized subscripts, lowercase non-subscripts). Also, Ho is sometimes H0 or HO.
Discussion 4.1 - This section could be broken up into 3 paragraphs.
Pakistan is typoed “Paskistan”.
4.2 – I don’t know what is meant by the phrase/sentence “comforting to the use of ecotype diffrence…”.
“The Wright’s F-statistic…” Which one? The reader shouldn’t have to go back and figure out Fst. Also, after this sentence the paragraph just dies. Some discussion as to what the comparison means is warranted instead of just listing others’ values. This same comment applies to all of the Discussion section. To me, the discussion does a lot of comparing to other indigenous populations, giving their values, but lacks detail in discussing the meaning of their own results to support the results and conclusions.
“Our findings were in line with those reported by names…” I think this was an oversight in plugging in the names. However, throughout the Discussion, instead of names of authors, a reference number is referred to, and to me, it breaks up the flow and readability. If you don’t want to use other authors’ names, the way you occasionally phrase sentences (i.e. 4.3 “Similar results were observed in Asian and Vietnamese local chicken populations [39].”) and how you wrote the Introduction has a much better flow. Side note – isn’t Vietnam in Asia?
The Conclusion section is concise, but how you came to the conclusions can use more discussion in the Discussion section.
Reviewer 2 Report
The authors investigate the genetic diversity and population structure of local chickens breeds with 20 microsatellite markers. Although using microsatellite markers to estimate the genetic diversity is recommended by FAO, it can only provide limit information relative to whole-genome SNP. The content is lack of novelty. Several concerns were listed as follows.
1. Why you choose to use 20 markers instead of 30 markers?
2. You can compare the genetic diversity or popultion structure using both microsatellite markers and SNP.
3. Gene flow analysis can be performed.
4. It's better to focus on the germpalsm characterisitics of some breed instead of the simple population structure analysis.
Reviewer 3 Report
The authors try to investigate the genetic diversity and population structure of local chicken ecotypes from Burkina Faso using microsatellite markers. A total of 71 individuals representing four local chicken populations were studied. Their results showed that local chickens in Burkina Faso have a rich genetic diversity with little differentiation between the studied populations. This study provides important information on measures of genetic diversity that could help in the design, and implementation of future genetic improvement and conservation programs for local chickens in Burkina Faso. Some minor corrections should be conducted.
1. Line 9 in 2.1.Please check the storage conditions of DNA.
2. Line 14-17 in 2.2.Change 94°,72°and 90°to 94℃ ,72℃ and 92℃.
3. Please indicate the meaning of “N” in the table.
4. Format the headings in 3.4 in the same way as the rest of the headings.
5. Modify the format of table 2.
6. Clearly indicate the meaning of “Ne” and “I” in Table 2.
7. Figure 2 is a little messy, please revise it again.
8. The text in the figures are too small, resize the figures, especially in figure 3.
